# A New Reverse Extended Hardy–Hilbert's Inequality with Two Partial Sums and Parameters

## Jianquan Liao and Bicheng Yang *

School of Mathematics, Guangdong University of Education, Guangzhou 510303, China; liaojianquan@gdei.edu.cn
* Correspondence: bcyang818@163.com

**Abstract:** By using the methods of real analysis and the mid-value theorem, we introduce some lemmas and obtain a new reverse extended Hardy–Hilbert's inequality with two partial sums and multi-parameters. We also give a few equivalent conditions of the best possible constant factor related to several parameters in the new inequality. Some particular inequalities are deduced.

**Keywords:** constant factor; mid-value theorem; Bernoulli function; parameter; reverse

**MSC:** 26D15





## 1. Introduction

If $p > 1$, $\frac{1}{p} + \frac{1}{q} = 1$, $a_m, b_n \geq 0$, $0 < \sum_{m=1}^{\infty} a_m^p < \infty$ and $0 < \sum_{n=1}^{\infty} b_n^q < \infty$, then we have the following well-known Hardy–Hilbert's inequality (cf. [1], Theorem 315):

$$\sum_{m=1}^{\infty} \sum_{n=1}^{\infty} \frac{a_m b_n}{m+n} < \frac{\pi}{\sin(\pi/p)} \left( \sum_{m=1}^{\infty} a_m^p \right)^{\frac{1}{p}} \left( \sum_{n=1}^{\infty} b_n^q \right)^{\frac{1}{q}}, \tag{1}$$

where the constant factor $\frac{\pi}{\sin(\pi/p)}$ is best possible.

In 2006, Krnic et al. [2] obtained the following inequality, which is a generalization of (1): for $\lambda_i \in (0, 2] (i = 1, 2)$, $\lambda_1 + \lambda_2 = \lambda \in (0, 4]$,

$$\sum_{m=1}^{\infty} \sum_{n=1}^{\infty} \frac{a_m b_n}{(m+n)^{\lambda}} < B(\lambda_1, \lambda_2) \left[ \sum_{m=1}^{\infty} m^{p(1-\lambda_1)-1} a_m^p \right]^{\frac{1}{p}} \left[ \sum_{n=1}^{\infty} n^{q(1-\lambda_2)-1} b_n^q \right]^{\frac{1}{q}}, \tag{2}$$

with the best possible constant factor $B(\lambda_1, \lambda_2)$, in which

$$B(u, v) = \int_0^{\infty} \frac{t^{u-1}}{(1+t)^{u+v}} \, dt (u, v > 0) \tag{3}$$

is the beta function. In particular, for $p = q = 2$, $\lambda_1 = \lambda_2 = \frac{\lambda}{2}$, (2) deduces to Yang's inequality in [3]. In 2019, following the way of (2), by using Abel's summation by parts formula, Adiyasuren et al. [4] provided the following extension of (2) involving two partial sums and some parameters, for

$$\sum_{m=1}^{\infty} \sum_{n=1}^{\infty} \frac{a_m b_n}{(m+n)^{\lambda}} < \lambda_1 \lambda_2 B(\lambda_1, \lambda_2) \left( \sum_{m=1}^{\infty} m^{-p\lambda_1 - 1} A_m^p \right)^{\frac{1}{p}} \left( \sum_{n=1}^{\infty} n^{-q\lambda_2 - 1} B_n^q \right)^{\frac{1}{q}}, \tag{4}$$

where $\lambda_1\lambda_2 B(\lambda_1, \lambda_2)$ is the best possible constant factor, and $A_m := \sum_{i=1}^m a_i$ and $B_n := \sum_{k=1}^n b_k$ $(m, n \in \mathrm{N} = \{1, 2, \cdots\})$, satisfy the following inequalities:

$$0 < \sum_{m=1}^{\infty} m^{-p\lambda_1-1} A_m^p < \infty \text{ and } 0 < \sum_{n=1}^{\infty} n^{-q\lambda_2-1} B_n^q < \infty.$$

Both (1) and (2) with their integral analogues played an important role in real analysis, in which some generalizations of (1) are given and a relation between (1) and the other Hilbert-type inequality is obtained (cf. [5–16]). In 2021, Gu et. al. [17] provided a generalization of (4) with $\frac{1}{(m^\alpha + n^\beta)^\lambda}$ $(\alpha, \beta \in (0, 1])$ as the kernel of inequality. In 2016, by using the weight coefficients and the techniques of real analysis, Hong et al. [18] considered a few equivalent statements of the generalization of (1) with the best possible constant factor related to multi-parameters. The other further results were obtained by [19–29].

In this article, based on the idea of [17,18], by using the techniques of real analysis and the mid-value theorem, we introduce some preserving lemmas and then give a reverse of (2) with two partial sums and multi-parameters, which is a new reverse version of the inequality in [16]. We also consider a few equivalent conditions of the best possible constant factor in the reverse inequality related to multi-parameters. Furthermore, several inequalities are deduced by setting some particular parameters.

## 2. Some Lemmas

In what follows, we assume that $0 < p < 1$ $(q < 0)$, $\frac{1}{p} + \frac{1}{q} = 1$, $\lambda \in (0, 6]$, $\alpha, \beta \in (0, 1]$, $\lambda_1 \in (0, \frac{2}{\alpha}] \cap (0, \lambda)$, $\lambda_2 \in (0, \frac{2}{\beta}] \cap (0, \lambda)$, $\hat{\lambda}_1 := \frac{\lambda - \lambda_2}{p} + \frac{\lambda_1}{q}$, $\hat{\lambda}_2 := \frac{\lambda - \lambda_1}{q} + \frac{\lambda_2}{p}$. We also assume that $a_m, b_n \geq 0$, $A_m := \sum_{j=1}^m a_j$, $B_n := \sum_{k=1}^n b_k$ $(m, n \in \mathrm{N})$, satisfy $A_m = o(e^{tm^\alpha})$, $B_n = o(e^{tn^\beta})$ $(t > 0; m, n \to \infty)$, and the following inequalities:

$$0 < \sum_{m=1}^{\infty} m^{p(1-\alpha\hat{\lambda}_1)-1} a_m^p < \infty, 0 < \sum_{n=1}^{\infty} n^{q(1-\beta\hat{\lambda}_2)-1} b_n^q < \infty. \tag{5}$$

**Lemma 1.** *For $t > 0$, the following inequalities on the partial sums are valid:*

$$\sum_{m=1}^{\infty} e^{-tm^\alpha} m^{\alpha-1} A_m \geq \frac{1}{t\alpha} \sum_{m=1}^{\infty} e^{-tm^\alpha} a_m, \tag{6}$$

$$\sum_{n=1}^{\infty} e^{-tn^\beta} n^{\beta-1} B_n \geq \frac{1}{t\beta} \sum_{n=1}^{\infty} e^{-tn^\beta} b_n. \tag{7}$$

**Proof.** Since $A_m e^{-tm^\alpha} = o(1)(m \to \infty)$, by Abel's summation by parts formula, it follows that

$$\sum_{m=1}^{\infty} e^{-tm^\alpha} a_m = \lim_{m \to \infty} A_m e^{-tm^\alpha} + \sum_{m=1}^{\infty} A_m \left[ e^{-tm^\alpha} - e^{-t(m+1)^\alpha} \right]$$
$$= \sum_{m=1}^{\infty} A_m \left[ e^{-tm^\alpha} - e^{-t(m+1)^\alpha} \right].$$

We set function $g(x) = e^{-tx^\alpha}, x \in [m, m+1]$. Then, we find $g'(x) = -t\alpha x^{\alpha-1}e^{-tx^\alpha}$, and for $\alpha \in (0,1]$, $h(x) := x^{\alpha-1}e^{-tx^\alpha}$ is decreasing in $[m, m+1]$. By the mid-value theorem, we have

$$
\begin{aligned}
\sum_{m=1}^{\infty} e^{-tm^\alpha} a_m &= -\sum_{m=1}^{\infty} A_m(g(m+1) - g(m)) \\
&= -\sum_{m=1}^{\infty} A_m g\prime(m+\theta) = t\alpha \sum_{m=1}^{\infty} (m+\theta)^{\alpha-1} e^{-t(m+\theta)^\alpha} A_m \\
&\le t\alpha \sum_{m=1}^{\infty} m^{\alpha-1} e^{-tm^\alpha} A_m (\theta \in (0,1)).
\end{aligned}
$$

Hence, we have (6). In the same way, inequality (7) follows.
The lemma is proved.　$\square$

In the following lemma, for estimating the weight coefficient in Lemma 3, we introduce some results related to the Bernoulli functions and the related formulas.

**Lemma 2.** *(Ref. [5]'s section 2.2.3, [30]) (i) If $(-1)^i \frac{d^i}{dt^i} g(t) > 0$, $t \in [m, \infty)(m \in N)$ with $g^{(i)}(\infty) = 0$ $(i = 0, 1, 2, 3)$, $P_i(t)$, $B_i$ $(i \in N)$ are Bernoulli functions and Bernoulli numbers of $i$-order, then*

$$
\int_m^{\infty} P_{2k-1}(t) g(t) dt = -\varepsilon_k \frac{B_{2k}}{2k} g(m) (0 < \varepsilon_k < 1; k = 1, 2, \cdots). \tag{8}
$$

*In particular, for $k = 1$, since $B_2 = \frac{1}{6}$, we find*

$$
-\frac{1}{12} g(m) < \int_m^{\infty} P_1(t) g(t) dt < 0; \tag{9}
$$

*For $k = 2$, based on $B_4 = -\frac{1}{30}$, it follows that*

$$
0 < \int_m^{\infty} P_3(t) g(t) dt < \frac{1}{120} g(m). \tag{10}
$$

*(ii) (Ref. [5]'s section 2.2.3, [30]) Suppose that $f(t)(> 0) \in C^3[m, \infty)$, $f^{(i)}(\infty) = 0$ $(i = 0, 1, 2, 3)$. We have the following Euler–Maclaurin summation formula:*

$$
\sum_{k=m}^{\infty} f(k) = \int_m^{\infty} f(t) dt + \frac{1}{2} f(m) + \int_m^{\infty} P_1(t) f'(t) dt, \tag{11}
$$

$$
\int_m^{\infty} P_1(t) f'(t) dt = -\frac{1}{12} f'(m) + \frac{1}{6} \int_m^{\infty} P_3(t) f'''(t) dt. \tag{12}
$$

**Lemma 3.** *For $s \in (0, 6]$, $s_2 \in (0, \frac{2}{\beta}] \cap (0, s)$, $k_s(s_2) := B(s_2, s - s_2)$, the weight coefficient is defined as follows:*

$$
\varpi_s(s_2, m) := m^{\alpha(s-s_2)} \sum_{n=1}^{\infty} \frac{\beta n^{\beta s_2 - 1}}{(m^\alpha + n^\beta)^s} (m \in N). \tag{13}
$$

*Then, the following inequalities are valid:*

$$
0 < k_s(s_2)\left(1 - O\left(\frac{1}{m^{\alpha s_2}}\right)\right) < \varpi_s(s_2, m) < k_s(s_2)(m \in N). \tag{14}
$$

*where we indicate that $O\left(\frac{1}{m^{\alpha s_2}}\right) := \frac{1}{k_s(s_2)} \int_0^{\frac{1}{m^\alpha}} \frac{u^{s_2-1}}{(1+u)^s} du > 0$.*

**Proof.** For any $m \in \mathbb{N}$, the function $g(m,t)$ is defined by: $g(m,t) := \frac{\beta t^{\beta s_2 - 1}}{(m^\alpha + t^\beta)^s}$ $(t > 0)$. In view of (11), we have

$$
\begin{aligned}
\sum_{n=1}^{\infty} g(m,n) &= \int_1^\infty g(m,t)dt + \tfrac{1}{2}g(m,1) + \int_1^\infty P_1(t)g'(m,t)dt \\
&= \int_0^\infty g(m,t)dt - h(m),
\end{aligned}
$$

where we set $h(m) := \int_0^1 g(m,t)dt - \tfrac{1}{2}g(m,1) - \int_1^\infty P_1(t)g'(m,t)dt$.

We obtain $-\tfrac{1}{2}g(m,1) = \frac{-\beta}{2(m^\alpha+1)^s}$. By integration by parts, it follows that

$$
\begin{aligned}
\int_0^1 g(m,t)dt &= \beta \int_0^1 \frac{t^{\beta s_2 - 1}}{(m^\alpha + t^\beta)^s}dt \overset{u=t^\beta}{=} \int_0^1 \frac{u^{s_2 - 1}}{(m^\alpha + u)^s}du \\
&= \frac{1}{s_2}\int_0^1 \frac{du^{s_2}}{(m^\alpha + u)^s} = \frac{1}{s_2}\frac{u^{s_2}}{(m^\alpha + u)^s}\Big|_0^1 + \frac{s}{s_2}\int_0^1 \frac{u^{s_2}}{(m^\alpha + u)^{s+1}}du \\
&= \frac{1}{s_2}\frac{1}{(m^\alpha + 1)^s} + \frac{s}{s_2(s_2+1)}\int_0^1 \frac{du^{s_2+1}}{(m^\alpha + u)^{s+1}} \\
&> \frac{1}{s_2}\frac{1}{(m^\alpha + 1)^s} + \frac{s}{s_2(s_2+1)}\left[\frac{u^{s_2+1}}{(m^\alpha + u)^{s+1}}\right]_0^1 + \frac{s(s+1)}{s_2(s_2+1)(m^\alpha+1)^{s+2}}\int_0^1 u^{s_2+1}du \\
&= \frac{1}{s_2}\frac{1}{(m^\alpha + 1)^s} + \frac{\lambda}{s_2(s_2+1)}\frac{1}{(m^\alpha+1)^{s+1}} + \frac{s(s+1)}{s_2(s_2+1)(s_2+2)}\frac{1}{(m^\alpha+1)^{s+2}},
\end{aligned}
$$

$$
\begin{aligned}
-g'(m,t) &= -\frac{\beta(\beta s_2 - 1)t^{\beta s_2 - 2}}{(m^\alpha + t^\beta)^s} + \frac{\beta^2 s t^{\beta + \beta s_2 - 2}}{(m^\alpha + t^\beta)^{s+1}} \\
&= -\frac{\beta(\beta s_2 - 1)t^{\beta s_2 - 2}}{(m^\alpha + t^\beta)^s} + \frac{\beta^2 s(m^\alpha + t^\beta - m^\alpha)t^{\beta s_2 - 2}}{(m^\alpha + t^\beta)^{s+1}} = \frac{\beta(\beta s - \beta s_2 + 1)t^{\beta s_2 - 2}}{(m^\alpha + t^\beta)^s} - \frac{\beta^2 s m^\alpha t^{\beta s_2 - 2}}{(m^\alpha + t^\beta)^{s+1}},
\end{aligned}
$$

and for $0 < s_2 \le \frac{2}{\beta}$, $0 < \beta \le 1$, $s_2 < s \le 6$, it follows that

$$
(-1)^i \frac{d^i}{dt^i}\Big[\frac{t^{\beta s_2 - 2}}{(m^\alpha + t^\beta)^s}\Big] > 0, \quad (-1)^i \frac{d^i}{dt^i}\Big[\frac{t^{\beta s_2 - 2}}{(m^\alpha + t^\beta)^{s+1}}\Big] > 0 \quad (i = 0,1,2,3).
$$

By (9), (10), (11) and (12), we obtain

$$
\beta(\beta s - \beta s_2 + 1)\int_1^\infty P_1(t)\frac{t^{\beta s_2 - 2}}{(m^\alpha + t^\beta)^s}dt > -\frac{\beta(\beta s - \beta s_2 + 1)}{12(m^\alpha + 1)^s},
$$

$$
\begin{aligned}
&-\beta^2 m^\alpha s \int_1^\infty P_1(t)\frac{t^{\beta s_2 - 2}}{(m^\alpha + t^\beta)^{s+1}}dt \\
&= \frac{\beta^2 m^\alpha s}{12(m^\alpha + 1)^{s+1}} - \frac{\beta^2 m^\alpha s}{6}\int_1^\infty P_3(t)\left[\frac{t^{\beta s_2 - 2}}{(m^\alpha + t^\beta)^{s+1}}\right]'' dt \\
&> \frac{\beta^2 m^\alpha s}{12(m^\alpha + 1)^{s+1}} - \frac{\beta^2 m^\alpha s}{720}\left[\frac{t^{\beta s_2 - 2}}{(m^\alpha + t^\beta)^{s+1}}\right]''_{t=1} \\
&> \frac{\beta^2(m^\alpha + 1 - 1)s}{12(m^\alpha + 1)^{s+1}} - \frac{\beta^2(m^\alpha + 1)s}{720}\left[\frac{(s+1)(s+2)\beta^2}{(m^\alpha+1)^{s+3}} + \frac{\beta(s+1)(5-\beta-2\beta s_2)}{(m^\alpha+1)^{s+2}} + \frac{(2-\beta s_2)(3-\beta s_2)}{(m^\alpha+1)^{s+1}}\right] \\
&= \frac{\beta^2 s}{12(m^\alpha+1)^s} - \frac{\beta^2 s}{12(m^\alpha+1)^{s+1}} \\
&\quad - \frac{\beta^2 s}{720}\Big[\frac{(s+1)(s+2)\beta^2}{(m^\alpha+1)^{s+2}} + \frac{\beta(s+1)(5-\beta-2\beta s_2)}{(m^\alpha+1)^{s+1}} + \frac{(2-\beta s_2)(3-\beta s_2)}{(m^\alpha+1)^s}\Big].
\end{aligned}
$$

Hence, we have $h(m) > \frac{1}{(m^\alpha+1)^s}h_1 + \frac{\lambda}{(m^\alpha+1)^{s+1}}h_2 + \frac{s(s+1)}{(m^\alpha+1)^{s+2}}h_3$, where

$$
h_1 := \frac{1}{s_2} - \frac{\beta}{2} - \frac{\beta - \beta^2 s_2}{12} - \frac{\beta^2 s(2 - \beta s_2)(3 - \beta s_2)}{720},
$$

$$
h_2 := \frac{1}{s_2(s_2+1)} - \frac{\beta^2}{12} - \frac{\beta^3(s+1)(5 - \beta - 2\beta s_2)}{720},
$$

and $h_3 := \frac{1}{s_2(s_2+1)(s_2+2)} - \frac{\beta^4(s+2)}{720}$. We can find

$$h_1 \geq \frac{1}{s_2} - \frac{\beta}{2} - \frac{\beta - \beta^2 s_2}{12} - \frac{s\beta^2(2 - \beta s_2)(3 - \beta s_2)}{720} = \frac{g(s_2)}{720 s_2},$$

where the function $g(\sigma)(\sigma \in (0, \frac{2}{\beta}])$ is indicated by

$$g(\sigma) := 720 - (420\beta + 6s\beta^2)\sigma + (60\beta^2 + 5s\beta^3)\sigma^2 - s\beta^4\sigma^3.$$

For $\beta \in (0, 1]$, $s \in (0, 6]$, we obtain

$$\begin{aligned} g'(\sigma) &= -(420\beta + 6s\beta^2) + 2(60\beta^2 + 5s\beta^3)\sigma - 3\beta^4\sigma^2 \\ &\leq -420\beta - 6s\beta^2 + 2(60\beta^2 + 5s\beta^3)\tfrac{2}{\beta} = (14s\beta - 180)\beta < 0, \end{aligned}$$

and then it follows that $h_1 \geq \frac{g(s_2)}{720\, s_2} \geq \frac{g(2/\beta)}{720\, s_2} = \frac{1}{6\, s_2} > 0$. For $s_2 \in (0, \frac{2}{\beta}]$, we still find

$$h_2 > \frac{\beta^2}{6} - \frac{\beta^2}{12} - \frac{5(s+1)\beta^2}{720} = (\frac{1}{12} - \frac{s+1}{140})\beta^2 > 0,$$

and $h_3 \geq (\frac{1}{24} - \frac{s+2}{720})\beta^3 > 0 (s \in (0, 6])$.

Hence, it follows that $h(m) > 0$. Setting $t = m^{\frac{\alpha}{\beta}} u^{\frac{1}{\beta}}$, we have

$$\begin{aligned} \varpi_s(s_2, m) &= m^{\alpha(s-s_2)} \sum_{n=1}^{\infty} g(m, n) < m^{\alpha(s-s_2)} \int_0^{\infty} g(m, t)dt \\ &= \beta m^{\alpha(s-s_2)} \int_0^{\infty} \frac{t^{\beta s_2 - 1}dt}{(m^\alpha + t^\beta)^s} = \int_0^{\infty} \frac{u^{s_2-1}du}{(1+u)^s} = B(s_2, s - s_2) = k_s(s_2). \end{aligned}$$

On the other hand, in view of (11), we find

$$\begin{aligned} \sum_{n=1}^{\infty} g(m, n) &= \int_1^{\infty} g(m, t)dt + \tfrac{1}{2}g(m, 1) + \int_1^{\infty} P_1(t)g'(m, t)dt \\ &= \int_1^{\infty} g(m, t)dt + H(m), \end{aligned}$$

where we set $H(m) := \frac{1}{2}g(m, 1) + \int_1^{\infty} P_1(t)g'(m, t)dt$.

We find $\frac{1}{2}g(m, 1) = \frac{\beta}{2(m^\alpha + 1)^s}$, and

$$g'(m, t) = -\frac{\beta(\beta s - \beta s_2 + 1)t^{\beta s_2 - 2}}{(m^\alpha + t^\beta)^s} + \frac{\beta^2 s m^\alpha t^{\beta s_2 - 2}}{(m^\alpha + t^\beta)^{s+1}}.$$

For $s_2 \in (0, \frac{2}{\beta}] \cap (0, s)$, $0 < s \leq 6$, by (7), we find

$$-\beta(\beta s - \beta s_2 + 1) \int_1^{\infty} P_1(t) \frac{t^{\beta s_2 - 2}}{(m^\alpha + t^\beta)^s} dt > 0,$$

and

$$\beta^2 m^\alpha s \int_1^{\infty} P_1(t) \frac{t^{\beta s_2 - 2}}{(m^\alpha + t^\beta)^{s+1}} dt > -\frac{\beta^2 m^\alpha s}{12(m^\alpha + 1)^{s+1}} > -\frac{\beta^2 s}{12(m^\alpha + 1)^s}.$$

Hence, it follows that

$$H(m) > \frac{\beta}{2(m^\alpha + 1)^s} - \frac{\beta^2 s}{12(m^\alpha + 1)^s} \geq \frac{\beta}{2(m^\alpha + 1)^s} - \frac{6\beta}{12(m^\alpha + 1)^s} = 0.$$

Then, we have

$$
\begin{aligned}
\omega_s(\lambda_2, m) &= m^{\alpha(s-s_2)} \sum_{n=1}^{\infty} g(m,n) > m^{\alpha(s-s_2)} \int_1^{\infty} g(m,t)dt \\
&= m^{\alpha(s-s_2)} \int_0^{\infty} g(m,t)dt - m^{\alpha(s-s_2)} \int_0^1 g(m,t)dt \\
&= k_s(s_2)\left[1 - \frac{1}{k_s(s_2)} \int_0^{\frac{1}{m^{\alpha}}} \frac{u^{s_2-1}}{(1+u)^s}du\right] > 0,
\end{aligned}
$$

where we indicate that $O(\frac{1}{m^{\alpha s_2}}) = \frac{1}{k_s(s_2)} \int_0^{\frac{1}{m^{\alpha}}} \frac{u^{s_2-1}}{(1+u)^s}du$, satisfying

$$
0 < \int_0^{\frac{1}{m^{\alpha}}} \frac{u^{s_2-1}}{(1+u)^s}du < \int_0^{\frac{1}{m^{\alpha}}} u^{s_2-1}du = \frac{1}{s_2 m^{\alpha s_2}}.
$$

Therefore, inequalities (14) are valid.
The lemma is proved. $\square$

In view of Lemma 3, the key inequality is obtained as follows:

**Lemma 4.** *We have the reverse inequality, as follows:*

$$
\begin{aligned}
I_\lambda &:= \sum_{n=1}^{\infty} \sum_{m=1}^{\infty} \frac{a_m b_n}{(m^{\alpha}+n^{\beta})^{\lambda}} > \left(\frac{1}{\beta}k_\lambda(\lambda_2)\right)^{\frac{1}{p}}\left(\frac{1}{\alpha}k_\lambda(\lambda_1)\right)^{\frac{1}{q}} \\
&\times \left[\sum_{m=1}^{\infty}(1-O(\frac{1}{m^{\alpha\lambda_2}}))m^{p(1-\alpha\hat{\lambda}_1)-1}a_m^p\right]^{\frac{1}{p}}\left[\sum_{n=1}^{\infty}n^{q(1-\beta\hat{\lambda}_2)-1}b_n^q\right]^{\frac{1}{q}}.
\end{aligned}
\tag{15}
$$

**Proof.** By the symmetry, for $s_1 \in (0,\frac{2}{\alpha}] \cap (0,s), s \in (0,6]$, we obtain the inequalities of the next weight coefficient, as follows:

$$
\begin{aligned}
0 &< k_s(s_1)(1-O(\frac{1}{n^{\beta s_1}})) \\
&< \omega_s(s_1,n) := n^{\beta(s-s_1)} \sum_{m=1}^{\infty} \frac{\alpha m^{\alpha s_1-1}}{(m^{\alpha}+n^{\beta})^s} < k_s(s_1)(n \in \mathbb{N}),
\end{aligned}
\tag{16}
$$

where we indicate $O(\frac{1}{n^{\beta s_1}}) := \frac{1}{k_s(s_1)} \int_0^{\frac{1}{n^{\beta}}} \frac{u^{s_1-1}}{(1+u)^s}du > 0$.

Using the reverse Hölder's inequality (cf. [31]), it follows that

$$
\begin{aligned}
I_\lambda &= \sum_{n=1}^{\infty} \sum_{m=1}^{\infty} \frac{1}{(m^{\alpha}+n^{\beta})^{\lambda}}\left[\frac{m^{\alpha(1-\lambda_1)/q}(\beta n^{\beta-1})^{1/p}}{n^{\beta(1-\lambda_2)/p}(\alpha m^{\alpha-1})^{1/q}}a_m\right]\left[\frac{n^{\beta(1-\lambda_2)/p}(\alpha m^{\alpha-1})^{1/q}}{m^{\alpha(1-\lambda_1)/q}(\beta n^{\beta-1})^{1/p}}b_n\right] \\
&\geq \left[\sum_{m=1}^{\infty} \sum_{n=1}^{\infty} \frac{\beta}{(m^{\alpha}+n^{\beta})^{\lambda}} \frac{m^{\alpha(1-\lambda_1)(p-1)}n^{\beta-1}a_m^p}{n^{\beta(1-\lambda_2)}(\alpha m^{\alpha-1})^{p-1}}\right]^{\frac{1}{p}}\left[\sum_{n=1}^{\infty} \sum_{m=1}^{\infty} \frac{\alpha}{(m^{\alpha}+n^{\beta})^{\lambda}} \frac{n^{\beta(1-\lambda_2)(q-1)}m^{\alpha-1}b_n^q}{m^{\alpha(1-\lambda_1)}(\beta n^{\beta-1})^{q-1}}\right]^{\frac{1}{q}} \\
&= \left(\frac{1}{\alpha}\right)^{\frac{1}{q}}\left(\frac{1}{\beta}\right)^{\frac{1}{p}}\left[\sum_{m=1}^{\infty} \omega_\lambda(\lambda_2,m)m^{p(1-\alpha\hat{\lambda}_1)-1}a_m^p\right]^{\frac{1}{p}} \\
&\times \left[\sum_{n=1}^{\infty} \omega_\lambda(\lambda_1,n)n^{q(1-\beta\hat{\lambda}_2)-1}b_n^q\right]^{\frac{1}{q}}.
\end{aligned}
$$

By (14), (16) and (5) (for $s=\lambda, s_i=\lambda_i (i=1,2)$), since $0 < p < 1(q < 0)$ and the assumptions, we obtain (15).
The lemma is proved. $\square$

## 3. Main Results

By Lemma 1 and Lemma 4, the following theorem follows:

**Theorem 1.** *The following reverse inequality with two partial sums and parameters is valid:*

$$I := \sum_{m=1}^{\infty} \sum_{n=1}^{\infty} \frac{m^{\alpha-1} n^{\beta-1}}{(m^\alpha + n^\beta)^{\lambda+2}} A_m B_n > \frac{\Gamma(\lambda)}{\Gamma(\lambda+2)\alpha\beta} \left(\tfrac{1}{\beta} k_\lambda(\lambda_2)\right)^{\frac{1}{p}} \left(\tfrac{1}{\alpha} k_\lambda(\lambda_1)\right)^{\frac{1}{q}}$$

$$\times \left[ \sum_{m=1}^{\infty} (1 - O(\tfrac{1}{m^{\alpha\lambda_2}})) m^{p(1-\alpha\hat\lambda_1)-1} a_m^p \right]^{\frac{1}{p}} \left[ \sum_{n=1}^{\infty} n^{q(1-\beta\hat\lambda_2)-1} b_n^q \right]^{\frac{1}{q}}. \tag{17}$$

*In particular, for $\lambda_1 + \lambda_2 = \lambda$, we have*

$$0 < \sum_{m=1}^{\infty} m^{p(1-\alpha\lambda_1)-1} a_m^p < \infty, \ 0 < \sum_{n=1}^{\infty} n^{q(1-\beta\lambda_2)-1} b_n^q < \infty,$$

*as well as:*

$$\sum_{m=1}^{\infty} \sum_{n=1}^{\infty} \frac{m^{\alpha-1} n^{\beta-1} A_m B_n}{(m^\alpha + n^\beta)^{\lambda+2}} > \left(\frac{1}{\alpha}\right)^{1+\frac{1}{q}} \left(\frac{1}{\beta}\right)^{1+\frac{1}{p}} \frac{\Gamma(\lambda)}{\Gamma(\lambda+2)} B(\lambda_1, \lambda_2)$$

$$\times \left[ \sum_{m=1}^{\infty} (1 - O(\tfrac{1}{m^{\alpha\lambda_2}})) m^{p(1-\alpha\lambda_1)-1} a_m^p \right]^{\frac{1}{p}} \left[ \sum_{n=1}^{\infty} n^{q(1-\beta\lambda_2)-1} b_n^q \right]^{\frac{1}{q}}. \tag{18}$$

**Proof.** Based on the expression as follows

$$\frac{1}{(m^\alpha + n^\beta)^{\lambda+2}} = \frac{1}{\Gamma(\lambda+2)} \int_0^{\infty} t^{(\lambda+2)-1} e^{-(m^\alpha + n^\beta)t} dt,$$

by (6) and (7), we have

$$I = \frac{1}{\Gamma(\lambda+2)} \sum_{m=1}^{\infty} \sum_{n=1}^{\infty} (m^{\alpha-1} A_m)(n^{\beta-1} B_n) \int_0^{\infty} t^{\lambda+1} e^{-(m^\alpha + n^\beta)t} dt$$

$$= \frac{1}{\Gamma(\lambda+2)} \int_0^{\infty} t^{\lambda+1} \left( \sum_{m=1}^{\infty} e^{-m^\alpha t} m^{\alpha-1} A_m \right) \left( \sum_{n=1}^{\infty} e^{-n^\beta t} n^{\beta-1} B_n \right) dt$$

$$\geq \frac{1}{\Gamma(\lambda+2)} \int_0^{\infty} t^{\lambda+1} \left( \frac{1}{t\alpha} \sum_{m=1}^{\infty} e^{-m^\alpha t} a_m \right) \left( \frac{1}{t\beta} \sum_{n=1}^{\infty} e^{-n^\beta t} b_n \right) dt$$

$$= \frac{1}{\Gamma(\lambda+2)\alpha\beta} \sum_{m=1}^{\infty} \sum_{n=1}^{\infty} a_m b_n \int_0^{\infty} t^{\lambda-1} e^{-(m^\alpha + n^\beta)t} dt = \frac{\Gamma(\lambda)}{\Gamma(\lambda+2)\alpha\beta} I_\lambda.$$

Then, by (15), inequality (17) follows.
The theorem is proved. □

In the following two theorems, we provide a few equivalent conditions on (17).

**Theorem 2.** *Assume that $\lambda_1 \in (0, \frac{2}{\alpha} - 1] \cap (0, \lambda)$, $\lambda_2 \in (0, \frac{2}{\beta}) \cap (0, \lambda)$. $\lambda \in (0, 4]$. If $\lambda_1 + \lambda_2 = \lambda$, then $\frac{\Gamma(\lambda)}{\Gamma(\lambda+2)\alpha\beta} \left(\tfrac{1}{\beta} k_\lambda(\lambda_2)\right)^{\frac{1}{p}} \left(\tfrac{1}{\alpha} k_\lambda(\lambda_1)\right)^{\frac{1}{q}}$ in (17) is the best possible. constant factor.*

**Proof.** We now prove that the constant factor $\frac{\Gamma(\lambda)}{\Gamma(\lambda+2)} \left(\frac{1}{\alpha}\right)^{1+\frac{1}{q}} \left(\frac{1}{\beta}\right)^{1+\frac{1}{p}} B(\lambda_1, \lambda_2)$ in (18) is the best possible. For any $0 < \varepsilon < \min\{p\lambda_1, |q|(\frac{2}{\beta} - \lambda_2)\}$, we set

$$\widetilde{a}_m := m^{\alpha(\lambda_1 - \frac{\varepsilon}{p})-1}, \ \widetilde{b}_n := n^{\beta(\lambda_2 - \frac{\varepsilon}{q})-1} (m, n \in \mathbb{N}).$$

Since $0 < \lambda_1 - \frac{\varepsilon}{p} \leq \frac{2}{\alpha} - 1$, $0 < \alpha(\lambda_1 - \frac{\varepsilon}{p}) \leq 2 - \alpha < 2$, by (2.2.24) (cf. [5]), we have

$$\widetilde{A}_m := \sum_{i=1}^{m} \widetilde{a}_i = \sum_{i=1}^{m} i^{\alpha(\lambda_1 - \frac{\varepsilon}{p})-1} = \int_1^m t^{\alpha(\lambda_1 - \frac{\varepsilon}{p})-1} dt$$

$$+ \frac{1}{2} [m^{\alpha(\lambda_1 - \frac{\varepsilon}{p})-1} + 1] + \frac{\varepsilon_0}{12} [\alpha(\lambda_1 - \frac{\varepsilon}{p}) - 1][m^{\alpha(\lambda_1 - \frac{\varepsilon}{p})-2} - 1]$$

$$= \frac{1}{\alpha(\lambda_1 - \frac{\varepsilon}{p})} (m^{\alpha(\lambda_1 - \frac{\varepsilon}{p})} + c_1 + O_1(m^{\alpha(\lambda_1 - \frac{\varepsilon}{p})-1})(\varepsilon_0 \in (0,1); m \in \mathbb{N}, m \to \infty).$$

In the same way, for $0 < \beta(\lambda_2 - \frac{\varepsilon}{q}) < 2$, we have

$$\widetilde{B}_n := \sum_{k=1}^{n} \widetilde{b}_k = \frac{1}{\beta(\lambda_2 - \frac{\varepsilon}{q})}(n^{\beta(\lambda_2 - \frac{\varepsilon}{q})} + c_2 + O_2(n^{\beta(\lambda_2 - \frac{\varepsilon}{q})-1}))(n \in \mathrm{N}; n \to \infty).$$

We observe that $\widetilde{A}_m = o(e^{tm^{\alpha}}), \widetilde{B}_n = o(e^{tn^{\beta}}) \, (t > 0; m, n \to \infty).$

If there exists a constant $M \geq \left(\frac{1}{\alpha}\right)^{1+\frac{1}{q}}\left(\frac{1}{\beta}\right)^{1+\frac{1}{p}}\frac{\Gamma(\lambda)}{\Gamma(\lambda+2)}B(\lambda_1, \lambda_2)$, such that (18) is valid when we replace $\left(\frac{1}{\alpha}\right)^{1+\frac{1}{q}}\left(\frac{1}{\beta}\right)^{1+\frac{1}{p}}\frac{\Gamma(\lambda)}{\Gamma(\lambda+2)}B(\lambda_1, \lambda_2)$ by $M$, then in particular, we have

$$\begin{aligned}
\widetilde{I} &:= \sum_{n=1}^{\infty}\sum_{m=1}^{\infty} \frac{m^{\alpha-1}n^{\beta-1}}{(m^{\alpha}+n^{\beta})^{\lambda+2}}\widetilde{A}_m\widetilde{B}_n \\
&> M\left[\sum_{m=1}^{\infty}\left(1 - O\left(\frac{1}{m^{\alpha\lambda_2}}\right)\right)m^{p(1-\alpha\lambda_1)-1}\widetilde{a}_m^p\right]^{\frac{1}{p}}\left[\sum_{n=1}^{\infty}n^{q(1-\beta\lambda_2)-1}\widetilde{b}_n^q\right]^{\frac{1}{q}}.
\end{aligned} \tag{19}$$

By (19) and using the decreasingness property of series, it follows that

$$\begin{aligned}
\widetilde{I} &> M\left[\sum_{m=1}^{\infty}m^{-\alpha\varepsilon-1} - \sum_{m=1}^{\infty}m^{-\alpha\varepsilon-1}O\left(\frac{1}{m^{\varepsilon\lambda_2}}\right)\right]^{\frac{1}{p}}\left(1 + \sum_{n=2}^{\infty}n^{-\beta\varepsilon-1}\right)^{\frac{1}{q}} \\
&> M\left(\int_1^{\infty}x^{-\alpha\varepsilon-1}dx - O(1)\right)^{\frac{1}{p}}\left(1 + \int_1^{\infty}y^{-\beta\varepsilon-1}dy\right)^{\frac{1}{q}} \\
&= \frac{M}{\varepsilon}\left(\frac{1}{\alpha} - \varepsilon O(1)\right)^{\frac{1}{p}}\left(\varepsilon + \frac{1}{\beta}\right)^{\frac{1}{q}}.
\end{aligned}$$

We still find that

$$\begin{aligned}
\widetilde{I} &< \frac{1}{\alpha(\lambda_1 - \frac{\varepsilon}{p})}\frac{1}{\beta(\lambda_2 - \frac{\varepsilon}{q})}\sum_{n=1}^{\infty}\sum_{m=1}^{\infty}\frac{1}{(m^{\alpha}+n^{\beta})^{\lambda+2}}\left[m^{\alpha-1}\left(m^{\alpha(\lambda_1-\frac{\varepsilon}{p})} + |c_1| + |O_1(m^{\alpha(\lambda_1-\frac{\varepsilon}{p})-1})|\right)\right] \\
&\times \left[n^{\beta-1}\left(n^{\beta(\lambda_2-\frac{\varepsilon}{q})} + |c_2| + |O_2(n^{\beta(\lambda_2-\frac{\varepsilon}{q})-1})|\right)\right] = \frac{1}{\alpha(\lambda_1-\frac{\varepsilon}{p})}\frac{1}{\beta(\lambda_2-\frac{\varepsilon}{q})} \\
&\times \sum_{n=1}^{\infty}\sum_{m=1}^{\infty}\frac{1}{(m^{\alpha}+n^{\beta})^{\lambda+2}}\left[m^{\alpha(\lambda_1-\frac{\varepsilon}{p}+1)-1} + |c_1|m^{\alpha-1} + |O_1(m^{\alpha(\lambda_1-\frac{\varepsilon}{p}+1)-2})|\right] \\
&\times \left[n^{\beta(\lambda_2-\frac{\varepsilon}{q}+1)-1} + |c_2|n^{\beta-1} + |O_2(n^{\beta(\lambda_2-\frac{\varepsilon}{q}+1)-2})|\right] \\
&= \frac{1}{\alpha(\lambda_1-\frac{\varepsilon}{p})}\frac{1}{\beta(\lambda_2-\frac{\varepsilon}{q})}(I_0 + I_1),
\end{aligned}$$

where we indicate that $I_0 := \sum_{n=1}^{\infty}\left[n^{\beta(\lambda_2-\frac{\varepsilon}{q}+1)} - \sum_{m=1}^{\infty}\frac{m^{\alpha(\lambda_1-\frac{\varepsilon}{p}+1)-1}}{(m^{\alpha}+n^{\beta})^{\lambda+2}}\right]$, and

$$\begin{aligned}
I_1 &:= \sum_{n=1}^{\infty}\sum_{m=1}^{\infty}\frac{1}{(m^{\alpha}+n^{\beta})^{\lambda+2}}\left[(|c_1|m^{\alpha-1} + |O_1(m^{\alpha(\lambda_1-\frac{\varepsilon}{p}+1)-2})|)n^{\beta(\lambda_2-\frac{\varepsilon}{q}+1)-1}|\right. \\
&\quad + (|c_1|m^{\alpha-1} + |O_1(m^{\alpha(\lambda_1-\frac{\varepsilon}{p}+1)-2})|)(|c_2|n^{\beta-1} + |O_2(n^{\beta(\lambda_2-\frac{\varepsilon}{q}+1)-2})|) \\
&\quad \left. + m^{\alpha(\lambda_1-\frac{\varepsilon}{p}+1)-1}(|c_2|n^{\beta-1} + |O_2(n^{\beta(\lambda_2-\frac{\varepsilon}{q}+1)-2})|)\right] \\
&\leq \sum_{n=1}^{\infty}\frac{n^{\beta(\lambda_2-\frac{\varepsilon}{q}+1)-1}}{(n^{\beta})^{\lambda_2+1}}\sum_{m=1}^{\infty}\frac{1}{(m^{\alpha})^{\lambda_1+1}}(|c_1|m^{\alpha-1} + |O_1(m^{\alpha(\lambda_1-\frac{\varepsilon}{p}+1)-2})|) \\
&\quad + \sum_{n=1}^{\infty}\frac{(|c_2|n^{\beta-1}+|O_2(n^{\beta(\lambda_2-\frac{\varepsilon}{q}+1)-2})|)}{(n^{\beta})^{\lambda_2+1}}\sum_{m=1}^{\infty}\frac{1}{(m^{\alpha})^{\lambda_1+1}}(|c_1|m^{\alpha-1} + |O_1(m^{\alpha(\lambda_1-\frac{\varepsilon}{p}+1)-2}|) \\
&\quad + \sum_{n=1}^{\infty}\frac{(|c_2|n^{\beta-1}+|O_2(n^{\beta(\lambda_2-\frac{\varepsilon}{q}+1)-2})|)}{(n^{\beta})^{\lambda_2+1}}\sum_{m=1}^{\infty}\frac{1}{(m^{\alpha})^{\lambda_1+1}}m^{\alpha(\lambda_1-\frac{\varepsilon}{p}+1)-1} \leq M_1 < \infty.
\end{aligned}$$

By (14), for $s = \lambda + 2 \in (0,6], s_1 = \lambda_1 + 1 - \frac{\varepsilon}{p}$ ($\in (0, \frac{2}{\alpha}] \cap (0, \lambda + 2)$), we have

$$
\begin{aligned}
I_0 &= \frac{1}{\alpha} \sum_{n=1}^{\infty} \left[ n^{\beta(\lambda_2 + 1 + \frac{\varepsilon}{p})} \sum_{m=1}^{\infty} \frac{\alpha m^{\alpha(\lambda_1 + 1 - \frac{\varepsilon}{p}) - 1}}{(m+n)^{\lambda+2}} \right] n^{-\beta\varepsilon - 1} \\
&= \frac{1}{\alpha} \sum_{n=1}^{\infty} \omega_{\lambda+2}(\lambda_1 + 1 - \frac{\varepsilon}{p}, n) n^{-\beta\varepsilon - 1} \\
&< \frac{1}{\alpha} k_{\lambda+2}(\lambda_1 + 1 - \frac{\varepsilon}{p}) \left( 1 + \sum_{n=2}^{\infty} n^{-\beta\varepsilon - 1} \right) \\
&< \frac{1}{\alpha} k_{\lambda+2}(\lambda_1 + 1 - \frac{\varepsilon}{p}) \left( 1 + \int_1^{\infty} y^{-\beta\varepsilon - 1} dy \right) \\
&= \frac{1}{\varepsilon} \frac{1}{\alpha\beta} B(\lambda_1 + 1 - \frac{\varepsilon}{p}, \lambda_2 + 1 + \frac{\varepsilon}{p})(1 + \beta\varepsilon).
\end{aligned}
$$

Based on the above results, we have

$$
\frac{1}{\alpha^2 (\lambda_1 - \frac{\varepsilon}{p})} \frac{1}{\beta^2 (\lambda_2 - \frac{\varepsilon}{q})} [B(\lambda_1 + 1 - \frac{\varepsilon}{p}, \lambda_2 + 1 + \frac{\varepsilon}{p})(1 + \beta\varepsilon) + \varepsilon M_1] > \varepsilon \widetilde{I} > M \left( \frac{1}{\alpha} - \varepsilon O(1) \right)^{\frac{1}{p}} \left( \varepsilon + \frac{1}{\beta} \right)^{\frac{1}{q}}.
$$

Setting $\varepsilon \to 0^+$ in the above inequality, in virtue of the continuity of the beta function, we find

$$
\left( \frac{1}{\alpha} \right)^{1 + \frac{1}{q}} \left( \frac{1}{\beta} \right)^{1 + \frac{1}{p}} \frac{\Gamma(\lambda)}{\Gamma(\lambda + 2)} B(\lambda_1, \lambda_2) = \left( \frac{1}{\alpha} \right)^{1 + \frac{1}{q}} \left( \frac{1}{\beta} \right)^{1 + \frac{1}{p}} \frac{1}{\lambda_1 \lambda_2} B(\lambda_1 + 1, \lambda_2 + 1) \geq M.
$$

Hence, $M = \left( \frac{1}{\alpha} \right)^{1 + \frac{1}{q}} \left( \frac{1}{\beta} \right)^{1 + \frac{1}{p}} \frac{\Gamma(\lambda)}{\Gamma(\lambda+2)} B(\lambda_1, \lambda_2)$ is the best possible constant factor in (18).

The theorem is proved.   □

**Theorem 3.** *Suppose that $\lambda_1 \in (0, \frac{2}{\alpha}] \cap (0, \lambda)$, $\lambda_2 \in \left( 0, \frac{2}{\beta} \right] \cap (0, \lambda)$. $\lambda \in (0, 6]$. If the constant factor $\frac{\Gamma(\lambda)}{\Gamma(\lambda+2)\alpha\beta} \left( \frac{1}{\beta} k_\lambda(\lambda_2) \right)^{\frac{1}{p}} \left( \frac{1}{\alpha} k_\lambda(\lambda_1) \right)^{\frac{1}{q}}$ in (17) is the best possible, then for*

$$
\lambda - \lambda_1 - \lambda_2 \in (-p\lambda_1, p(\lambda - \lambda_1)) \cap [q(\frac{2}{\beta} - \lambda_2), p(\frac{2}{\alpha} - \lambda_1)],
$$

*we have $\lambda_1 + \lambda_2 = \lambda$.*

**Proof.** For $\hat{\lambda}_1 = \frac{\lambda - \lambda_2}{p} + \frac{\lambda_1}{q} = \frac{\lambda - \lambda_1 - \lambda_2}{p} + \lambda_1$, $\hat{\lambda}_2 = \frac{\lambda - \lambda_1}{q} + \frac{\lambda_2}{p} = \frac{\lambda - \lambda_1 - \lambda_2}{q} + \lambda_2$, we find $\hat{\lambda}_1 + \hat{\lambda}_2 = \lambda$. For $\lambda - \lambda_1 - \lambda_2 \in (-p\lambda_1, p(\lambda - \lambda_1))$, we have $\hat{\lambda}_1 \in (0, \lambda)$, and then $\hat{\lambda}_2 = \lambda - \hat{\lambda}_1 \in (0, \lambda)$; for $\lambda - \lambda_1 - \lambda_2 \in [q(\frac{2}{\beta} - \lambda_2), p(\frac{2}{\alpha} - \lambda_1)]$, we still have $\hat{\lambda}_1 \leq \frac{2}{\alpha}$, $\hat{\lambda}_2 \leq \frac{2}{\beta}$. Then, for $\lambda_1 + \lambda_2 = \lambda$ in (17), substitution of $\lambda_i = \hat{\lambda}_i$ $(i = 1, 2)$, we still have

$$
\begin{aligned}
\sum_{m=1}^{\infty} \sum_{n=1}^{\infty} \frac{A_m B_n}{(m^\alpha + n^\beta)^{\lambda+2}} &> \left( \frac{1}{\alpha} \right)^{1 + \frac{1}{q}} \left( \frac{1}{\beta} \right)^{1 + \frac{1}{p}} \frac{\Gamma(\lambda)}{\Gamma(\lambda+2)} B(\hat{\lambda}_1, \hat{\lambda}_2) \\
&\times \left[ \sum_{m=1}^{\infty} (1 - O(\frac{1}{m^{\alpha \hat{\lambda}_2}})) m^{p(1 - \alpha\hat{\lambda}_1) - 1} a_m^p \right]^{\frac{1}{p}} \left[ \sum_{n=1}^{\infty} n^{q(1 - \beta\hat{\lambda}_2) - 1} b_n^q \right]^{\frac{1}{q}}.
\end{aligned}
\tag{20}
$$

By using the reverse Hölder's inequality (cf. [31]), we still obtain

$$
\begin{aligned}
B(\hat{\lambda}_1, \hat{\lambda}_2) &= k_\lambda \left( \frac{\lambda - \lambda_2}{p} + \frac{\lambda_1}{q} \right) \\
&= \int_0^\infty \frac{1}{(1+u)^\lambda} u^{\frac{\lambda-\lambda_2}{p} + \frac{\lambda_1}{q} - 1} du = \int_0^\infty \frac{1}{(1+u)^\lambda} \left( u^{\frac{\lambda-\lambda_2-1}{p}} \right) \left( u^{\frac{\lambda_1-1}{q}} \right) du \\
&\geq \left[ \int_0^\infty \frac{1}{(1+u)^\lambda} u^{\lambda-\lambda_2-1} du \right]^{\frac{1}{p}} \left[ \int_0^\infty \frac{1}{(1+u)^\lambda} u^{\lambda_1-1} du \right]^{\frac{1}{q}} \\
&= \left[ \int_0^\infty \frac{1}{(1+v)^\lambda} v^{\lambda_2-1} dv \right]^{\frac{1}{p}} \left[ \int_0^\infty \frac{1}{(1+u)^\lambda} u^{\lambda_1-1} du \right]^{\frac{1}{q}} \\
&= (k_\lambda(\lambda_2))^{\frac{1}{p}} (k_\lambda(\lambda_1))^{\frac{1}{q}}.
\end{aligned}
\tag{21}
$$

If $\frac{\Gamma(\lambda)}{\Gamma(\lambda+2)\alpha\beta} \left( \frac{1}{\beta} k_\lambda(\lambda_2) \right)^{\frac{1}{p}} \left( \frac{1}{\alpha} k_\lambda(\lambda_1) \right)^{\frac{1}{q}}$ in (17) is the best possible constant factor, then compare it with the constant factors in (17) and (20), and we have the following inequality:

$$
\begin{aligned}
& \frac{\Gamma(\lambda)}{\Gamma(\lambda+2)\alpha\beta} \left( \frac{1}{\beta} k_\lambda(\lambda_2) \right)^{\frac{1}{p}} \left( \frac{1}{\alpha} k_\lambda(\lambda_1) \right)^{\frac{1}{q}} \\
& \geq \left( \frac{1}{\alpha} \right)^{1+\frac{1}{q}} \left( \frac{1}{\beta} \right)^{1+\frac{1}{p}} \frac{\Gamma(\lambda)}{\Gamma(\lambda+2)} B(\hat{\lambda}_1, \hat{\lambda}_2) (\in \mathbf{R}_+),
\end{aligned}
$$

namely, $B(\hat{\lambda}_1, \hat{\lambda}_2) \leq (k_\lambda(\lambda_2))^{\frac{1}{p}} (k_\lambda(\lambda_1))^{\frac{1}{q}}$. Then, by (21), we have

$$
B(\hat{\lambda}_1, \hat{\lambda}_{21}) = (k_\lambda(\lambda_2))^{\frac{1}{p}} (k_\lambda(\lambda_1))^{\frac{1}{q}},
$$

which follows that (21) protains the form of equality.

We observe that (21) protains the form of equality if and only if there exist $A$ and $B$, such that they are not both zero and (cf. [31]) $Au^{\lambda-\lambda_2-1} = Bu^{\lambda_1-1} a.e.$ in $\mathbf{R}_+$. Assume that $A \neq 0$. It follows that $u^{\lambda-\lambda_2-\lambda_1} = \frac{B}{A} a.e.$ in $\mathbf{R}_+$, and then $\lambda - \lambda_2 - \lambda_1 = 0$. Hence, we have $\lambda_1 + \lambda_2 = \lambda$.

The theorem is proved. □

**Remark 1.** *(i) For $\alpha = \beta = 1, \lambda \in (0,4], \lambda_1 \in (0,1] \cap (0,\lambda), \lambda_2 \in (0,2) \cap (0,\lambda)$ in (18), we have the following reverse inequality with $\frac{1}{\lambda(\lambda+1)} B(\lambda_1, \lambda_2)$ as the best possible constant factor:*

$$
\begin{aligned}
& \sum_{m=1}^\infty \sum_{n=1}^\infty \frac{A_m B_n}{(m+n)^{\lambda+2}} > \frac{1}{\lambda(\lambda+1)} B(\lambda_1, \lambda_2) \\
& \times \left[ \sum_{m=1}^\infty \left( 1 - O\left( \frac{1}{m^{\lambda_2}} \right) \right) m^{p(1-\lambda_1)-1} a_m^p \right]^{\frac{1}{p}} \left[ \sum_{n=1}^\infty n^{q(1-\lambda_2)-1} b_n^q \right]^{\frac{1}{q}}.
\end{aligned}
\tag{22}
$$

*(ii) For $\alpha = \beta = \frac{1}{2}, \lambda \in (0,4], \lambda_1 \in (0,3] \cap (0,\lambda), \lambda_2 \in (0,\lambda)$ in (18), we have the following reverse inequality with $\frac{8}{\lambda(\lambda+1)} B(\lambda_1, \lambda_2)$ as the best possible constant factor:*

$$
\begin{aligned}
& \sum_{m=1}^\infty \sum_{n=1}^\infty \frac{A_m B_n}{(\sqrt{m}+\sqrt{n})^{\lambda+2} \sqrt{mn}} > \frac{8}{\lambda(\lambda+1)} B(\lambda_1, \lambda_2) \\
& \times \left[ \sum_{m=1}^\infty \left( 1 - O\left( \frac{1}{m^{\lambda_2/2}} \right) \right) m^{p(1-\frac{\lambda_1}{2})-1} a_m^p \right]^{\frac{1}{p}} \left[ \sum_{n=1}^\infty n^{q(1-\frac{\lambda_2}{2})-1} b_n^q \right]^{\frac{1}{q}}.
\end{aligned}
\tag{23}
$$

## 4. Conclusions

In this article, by using the techniques of real analysis, the way of weight coefficients and the idea of introduced parameters, applying the mid-value theorem. We estimate some lemmas and obtain a new reverse extended inequality (2) with two partial sums and multi-parameters in Theorem 1. We consider a few equivalent statements of the best possible constant factor related to several parameters in Theorems 2 and 3. We also deduce

some inequalities for setting particular parameters in Remark 1. The theorems and lemmas in this paper provide a useful extensive account of this type of inequality. Further studies should be using the idea of this article to build some other kinds of Hilbert-type inequalities with partial sums and parameters.

**Author Contributions:** B.Y. carried out the mathematical studies, participated in the sequence alignment and drafted the manuscript. J.L. participated in the design of the study and performed the numerical analysis. All authors have read and agreed to the published version of the manuscript.

**Funding:** This work is supported by the National Natural Science Foundation (No. 61772140), the Key Construction Discipline Scientific Research Ability Promotion Project of Guangdong Province (No 2021ZDJS056) and the Guangzhou Base Applied Research Project (No. 20220101181-7).

**Institutional Review Board Statement:** Not applicable.

**Informed Consent Statement:** Not applicable.

**Data Availability Statement:** Not applicable.

**Acknowledgments:** The authors thank the referees for their useful proposals to revise this paper.

**Conflicts of Interest:** The authors have no conflict of interest.

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
