# Peer review of "A New Reverse Extended Hardy–Hilbert’s Inequality with Two Partial Sums and Parameters"

_axioms, doi:10.3390/axioms12070678_

Round 1

Reviewer 1 Report

Review for the article "A New Reverse Extended Hardy-Hilbert’s Inequality with Two Partial Sums and Parameters" with ID axioms-2477450

Please see the following items.

1. The article is given without use of Axioms TeX style. It is needed to involve the journal's style and to format References according to the journal's style.

2. At the web page https://www.mdpi.com/journal/axioms/special_issues/799ES6D3GU , I see the title "Advances in Analysis and Control of Systems with Uncertainties II" and the key words of the special issue. Also I see that this issue belongs to the section "Mathematical Analysis". From the point of view of mathematical analysis, this article looks interesting. This article looks exclusively on mathematical analysis, but the special issue is devoted to analysis and control of systems with uncertainties. What do you think about this? 

3. Page 2. See the sentence "Inequality (1) with the integral analogues played an important role in analysis and its applications (cf. [5-15])." It is recommended to add something certain in Introduction about such inequalities' applications. Moreover, what do the authors think about possible applications of their results described in this article? 

3. I have found the article [Bicheng Yang, Lokenath Debnath, "On the extended Hardy–Hilbert's inequality", Journal of Mathematical Analysis and Applications, Volume 272, Issue 1, 2002, pages 187--199, https://doi.org/10.1016/S0022-247X(02)00151-8]. It is recommended to take this publication into account in References and add the corresponding comparing comment above with respect to the article submitted to Axioms.

4. It is recommended to check the whole article with respect to typos. E.g., Abstract is finished as follows: "some particular inequalities are. deduced". In the first line of Introduction, I see the word "and" in italic style. At the end of the page 1, I see "as follows: For" where the colon and the big letter "F" do not match. At the 2nd page, I see "Hong et al." where I do not see ",". And so on.

5. Page 2, Lemma 1. I see "Bernoulli functions and Bernoulli numbers". It is recommended to add some cite to some relevant book(s), because some readers (students) would see such cites. 

Reviewer 2 Report

The paper sound very interesting from a theoretical mathematical point of view. The authors should revise the paper following the comments below: 

- The abstract should be rewritten. The aim of the abstract section is to present the research question and the results of the research. Finally, the different methods to achieve that results. 

- The introduction section must be revised extensively. It looks like a mathematical formulation section. The aim of the introduction section is to present the relevant literature related to the current research.  

- Nomenclature MUST be added to the paper at the beginning of the paper. 

- The following paper can be added to the current research:

1:  Luo, R., Yang, B., & Huang, X. (2023). A reverse extended Hardy–Hilbert’s inequality with parameters. In Journal of Inequalities and Applications (Vol. 2023, Issue 1). Springer Science and Business Media LLC. https://doi.org/10.1186/s13660-023-02967-5

2:Nave, Op. (2020). Modification of Semi-Analytical Method Applied System of ODE. In Modern Applied Science (Vol. 14, Issue 6, p. 75). Canadian Center of Science and Education. https://doi.org/10.5539/mas.v14n6p75

 -  The authors should explain the connection between the presented Lemmas in section 2 and the results presented in section 3. 

- The authors must add a new section called discussion. In this section, the authors should explain in detail the theory and the results. 

- The conclusion section should be extended. 

Reviewer 3 Report

Paper axioms-2477450 “A New Reverse Extended Hardy-Hilbert’s Inequality with Two Partial Sums and Parameters”

Comments

This study investigates a new reverse extended Hardy-Hilbert’s inequality with two partial sums and parameters. I think the paper fits well the scope of the journal and addresses an important subject. However, a number of revisions are required before the paper can be considered for publication. There are some weak points that have to be strengthened. Below please find more specific comments:

*Abstract: The abstract should be expanded. I particular, I suggest adding a couple of sentences highlighting the contributions of this work to the state of the art and the major potential implications from the conducted research.

*Please check the manuscript for typos. I see that you have “inequalities are. deduced” in the abstract, but the period should be placed after “deduced”.

*Keywords: The keywords seem to be adequate. No comments.

*The introduction section should be improved. In particular, I suggest starting the introduction section verbally describing some preliminaries regarding the primary subject at hand instead of delving into many equations right away. This will certainly help the future readers to digest the information presented in this manuscript.

*Literature review: The manuscript does not have a literature review section and does not provide a thorough review of the relevant research works in this area. It is essential to review the relevant works that have been conducted by the researchers in this area, summarize the state of the art, identify the critical research gaps, and clearly explain the future readers how these gaps are addressed in the present research. A solid literature review will certainly strengthen the presentation of this manuscript. Otherwise, some readers may think that the manuscript is not complete.

*The manuscript has quite a lot of lemmas, propositions, theorems, etc. I suggest the authors to identify the most important lemmas, propositions, and theorems and keep them within the main body of the manuscript. All auxiliary lemmas, propositions, and theorems could be placed in an appendix. Some readers may be overwhelmed with the amount of lemmas, propositions, and theorems presented in the current version of the manuscript.

*Conclusions: The conclusion section should be more detailed. The author should not only discuss the outcomes of the present research but also discuss some future research needs. I suggest listing the future research needs using a set of bullet points. A more solid conclusion section would certainly improve the presentation of this manuscript as well.

n/a

Reviewer 4 Report

Using the weight coefficients, the Euler-Maclaurin summation formula, Abel’s summation by parts formula, and the mid-value theorem, a new reverse extended Hardy -Hilbert’s inequality with two partial sums and parameters is obtained . The equivalent statements of the best possible constant factor related to several parameters in the inequality are provided, and some particular inequalities are  derived.

I have some remarkss on exhibition of the paper.

1. Section 2  begins with  ... we assume that  ...(q<0 ) ?

2. Further ,in Lemma1 ,q=1,2,... ? 

 3. are. deduced   sould be   are  deduced many others errors of this type.

 4. Integration by parts, we find should be  By  Integration by parts, we find

So, the standard of the exposition/English, which will, in my view, need to be improved before the manuscript can be considered for publication.

Using the weight coefficients, the Euler-Maclaurin summation formula, Abel’s summation by parts formula, and the mid-value theorem, a new reverse extended Hardy -Hilbert’s inequality with two partial sums and parameters is obtained . The equivalent statements of the best possible constant factor related to several parameters in the inequality are provided, and some particular inequalities are  derived.

I have some remarkss on exhibition of the paper.

1. Section 2  begins with  ... we assume that  ...(q<0 ) ?

2. Further ,in Lemma1 ,q=1,2,... ?

 3. are. deduced   sould be   are  deduced many others errors of this type.

 4. Integration by parts, we find should be  By  Integration by parts, we find

So, the standard of the exposition/English, which will, in my view, need to be improved before the manuscript can be considered for publication.

Round 2

Reviewer 1 Report

Review (Round 2) for the article "A New Reverse Extended Hardy-Hilbert’s Inequality with Two Partial Sums and Parameters" with ID axioms-2477450

1. Page 2. See "Inequalities (1) and (2) with the integral analogues played an important role in analysis and its applications (cf. [5-16])". It is recommended to comment clearly what do you mean writing the word "applications" here. 

2. The article is still not designed taking into account the style TeX-files of Axioms.

3. Since Round 1, I note that this article looks exclusively on mathematical analysis, but the title and key words of this  special issue tell that the issue is conceived for analysis and control of systems with uncertainties (special issue "Advances in Analysis and Control of Systems with Uncertainties II",  https://www.mdpi.com/journal/axioms/special_issues/799ES6D3GU ). However, taking into account that the section is "Mathematical Analysis", taking into account that this article looks interesting with respect to mathematical analysis, taking into account the other reviews for this article, now I select "Accept after minor revision" and I think it's up to the editors to decide whether it is possible to take this article in this issue. 

Reviewer 3 Report

The authors took seriously my previous comments and made the required revisions in the manuscript. The quality and presentation of the manuscript have been improved. Therefore, I recommend acceptance.
